# Delivering Palliative and Supportive Care for Older Adults with Cancer: Interactions between Palliative Medicine and Geriatrics

**DOI:** 10.3390/cancers15153858

**Published:** 2023-07-29

**Authors:** Alicia Castelo-Loureiro, Andrea Perez-de-Acha, Ana Cristina Torres-Perez, Vanessa Cunha, Paola García-Valdés, Paula Cárdenas-Reyes, Enrique Soto-Perez-de-Celis

**Affiliations:** 1Department of Medical Oncology, Hospital 12 de Octubre, 28041 Madrid, Spain; aliciacastelo@hotmail.com; 2Department of Geriatrics, Instituto Nacional de Ciencias Médicas y Nutrición Salvador Zubirán, Mexico City 14080, Mexicoana.torresp@incmnsz.mx (A.C.T.-P.); pagava88@gmail.com (P.G.-V.);; 3School of Medicine, University of Toronto, Toronto, ON M5S 3G5, Canada; 4Department of Palliative Care, Hospital Gea González, Mexico City 14080, Mexico

**Keywords:** cancer, older adults, ageism, palliative care, pain, decision making

## Abstract

**Simple Summary:**

Advanced cancer represents the most common reason for needing palliative care among older adults worldwide. Unfortunately, older adults face several barriers when trying to obtain high-quality palliative care partly due to a lack of data regarding the best way to treat them and to limited geriatric expertise among clinicians. Providing palliative care for older patients with cancer requires considering the physiological and functional changes related with aging to tailor therapy and provide individualized care. Furthermore, since geriatrics and palliative care share many common goals, there is a need to create models of care that successfully integrate both disciplines. In this review, we summarize the current evidence regarding the provision of palliative care and symptom control for older adults, as well as guidance regarding the use of geriatrics for decision-making in older patients with incurable diseases.

**Abstract:**

The world’s population is aging rapidly, with projections indicating that by 2050 one in six people will be aged ≥65 years. As a result, the number of cancer cases in older people is expected to increase significantly. Palliative care is an essential component of cancer care with a direct impact on quality of life. However, older adults with cancer often suffer from multiple comorbidities, cognitive impairment, and frailty, posing unique challenges in the delivery of palliative care. The complex healthcare needs of older patients with cancer therefore require a comprehensive assessment, including a geriatric evaluation. Collaboration between geriatrics and palliative care can offer a solution to the challenges faced by older people with cancer, since this is a population with overlapping concerns for both disciplines. This review highlights the importance of palliative care for older adults with cancer and the benefits of a multidisciplinary approach. It also addresses the coordination of palliative care and geriatrics for specific symptom management and decision making.

## 1. Introduction

The world population is aging, and by 2050, one in six people, or 16% of the world, will be aged ≥65 years [1]. At the biological level, aging results from the accumulation of molecular and cellular damage, which in turn leads to a growing risk of cancer and other chronic diseases [2]. According to GLOBOCAN, the number of cancer cases in older individuals will rise from 9.95 million per year in 2020 to 18.6 million in 2040 [3]. Therefore, the need to provide high-quality care to older adults with cancer will increase in the coming years. People with cancer comprise approximately 34% of the population in need of supportive care worldwide. According to the World Health Organization (WHO), 56.8 million people need palliative care each year (25.7 million during the last year of life), of which approximately 40% are aged ≥70 years [4]. Unfortunately, currently only about 14% of those who need palliative care receive it, and this problem may become even worse due to population aging [5].

The need for palliative care tends to be greater among older adults because tumors are diagnosed at more advanced stages and often receive less oncological treatment [6,7]. Early palliative and supportive care are important for older patients with cancer, and should be part of the standard oncological care, regardless of disease stage. Palliative care can improve rates of survival and quality of life (QoL) as well as reduce the cost of treatment [8]. In addition, in some types of cancer, early integration of palliative care has been shown to have similar effects to first-line chemotherapy [8]. In older patients, the integration of geriatric medicine and palliative care into single multidisciplinary teams may represent a way to provide high-quality, tailored care [9]. This would unify competencies from geriatric medicine, oncology, and palliative care to respond to the socio-demographic changes and challenges of older adults with severe and life-limiting conditions, considering the particularities of this population (Figure 1) [9].

## 2. Aging-Related Changes and Palliative Care

Older adults may present with a different array of non-cancer-related problems than their younger counterparts, making palliative care more challenging. Often, older adults have multiple comorbidities, cognitive impairment, and frailty, which further complicates management [10,11]. Older patients with cancer are more likely to have geriatric syndromes than those without a cancer diagnosis [12]. This is because there are shared risk factors for geriatric syndromes and cancer, such as advanced age, cognitive impairment, functional disability, and limited mobility. Geriatric syndromes in patients with cancer are associated with increased risk of morbidity, mortality, adverse outcomes, and worse QoL [13].

In older patients, the accumulation of multiple chronic diseases (including cancer) and geriatric syndromes may lead to gradual deterioration, disability, and eventually death. This is often accompanied by multiple, often unpredictable, disease exacerbations, which need close medical supervision [14]. When providing palliative and supportive care for older adults with cancer, changes in pharmacodynamics and pharmacokinetics must be considered [12]. Likewise, limited organ function (such as chronic kidney disease), polypharmacy, and comorbidity, make older patients more vulnerable to adverse effects from cancer treatment [15]. This complexity explains why palliative and supportive care management requires a systematic evaluation of an older patient’s symptoms, health status, functionality, and geriatric syndromes or conditions.

This evaluation may be achieved through a geriatric assessment (GA), which includes validated tools to assess specific geriatric domains, which are known to be important in the management of older adults with cancer [16]. A GA includes an evaluation of comorbidities, screening for cognitive impairment and delirium, assessment of polypharmacy, functional status, physical performance (mobility), nutritional status, mental health, falls, and screening for social problems (environment, resources, and social support) [13,17]. Given that aging is multidimensional and highly heterogeneous, GA-guided interventions may be particularly useful for the creation of personalized plans aimed at managing older adults receiving palliative and supportive care.

## 3. The Intersection of Palliative Care and Geriatrics

‘Is geriatrics primarily palliative medicine or is palliative medicine primarily geriatrics?’ Although the fields of palliative medicine and geriatrics have developed from separate origins, they share much in common. Commonalities include goal-oriented care based on individual preferences; team-based interprofessional models of care; proactive multidimensional assessment and identification of unmet needs; adoption of the biopsychosocial model of care and attention to psychosocial factors; attention to caregiver needs and inclusion of caregivers in care planning and implementation; and value-added service to particularly vulnerable and frail older adults [18]. The goal of both specialties is to improve the QoL of patients and caregivers [18].

Based on their areas of synergy, many professionals have advocated the need for closer interaction and integration between geriatrics and palliative care. Presumably, collaborative working can foster high-quality care for older adults. Depending on their own strengths and areas of expertise, each specialty can contribute in different ways. Palliative care can contribute expertise in complex symptom management, ethical decision support, goal setting and advance care planning, and prognostication. Conversely, geriatrics can contribute expertise in GA, management of frailty, multimorbidity and polypharmacy, assessment of functional status and strategies to improve it, identification and management of cognitive impairment, and complex care coordination. From the collaboration between these two specialties emerges a new area: geriatric palliative care, unifying their competencies to address the growing need for better care of older patients with advanced life-limiting illness [19,20].

The importance of the integration and collaboration between geriatrics and palliative care has been recognized by the American Geriatrics Society and the American Academy of Hospice and Palliative Medicine. In 2009, both institutions started a collaborative effort to identify overlapping strengths and interests, and developed a joint proposal for ongoing collaboration in the areas of clinical care, leadership and organizational structure, research, education and training, and public policy (Table 1) [21].

While there are opportunities for integration between the two specialties, there are also barriers that need to be addressed. The boundaries between geriatrics and palliative care are unclear, so clarification of responsibilities within and between teams is mandatory, and improved communication is crucial [22].

Another challenge is that the evidence base for geriatric palliative care is sparse. Most studies tend to exclude older adults, as well as patients who are frail, have multimorbidity, or cognitive impairment, restricting the applicability of their findings. Issues such as the treatment of pain in older patients and its interaction with geriatric syndromes or the psychosocial and spiritual well-being of older populations have been vastly understudied, limiting the availability of tailored strategies to address them [23].

Care coordination may also be challenging since older patients usually have a prolonged illness course and transit thorough various healthcare settings. A lack of care coordination can be detrimental to older adults and decrease their QoL, given the higher risk of polypharmacy, conflicting recommendations, burdensome interventions, multiple visits to the emergency department, and recurrent hospital admissions [19]. The implementation of coordinated and continuous care for older adults needing palliative care is extremely important [19].

## 4. Management of Specific Symptoms in Older Adults with Cancer

### 4.1. Pain

Between 25% and 40% of older patients with cancer experience pain daily [24]. Pain is associated with increased dependence on activities of daily living, risk of falls, malnutrition, limitation in social activity, and increased risk of depression [25,26,27]. Since the etiology of pain in older adults with cancer is multifactorial, they benefit from a multidisciplinary team approach, including a geriatrician, a palliative care physician, a physical medicine and rehabilitation physician, physical therapists, and a psychiatrist or psychologist when appropriate [25,26,27,28]. The evaluation of pain in older adults may be a challenge since some (with and without cognitive impairment) may have difficulty responding to numerical pain scales. Other verbal descriptor scales, pain thermometers, and facial pain scales may have greater validity in older populations [25,27,29].

Treatment plans for older patients with cancer should incorporate non-pharmacological interventions such as massage, relaxation techniques, exercise, and rehabilitation [26,30,31]. Acetaminophen should be considered as a first-line treatment for pain management due to its proven efficacy and good safety profile [30]. Non-steroidal anti-inflammatory drugs should be administered at the lowest dose, for the shortest possible time, while monitoring for possible adverse effects (AEs) [30]. Although older people are more sensitive to the analgesic properties of opioids, they are also at higher risk of toxicity, and require close monitoring for AEs and adequate laxative therapy [30]. A good strategy is to use low-dose drug combinations (in which each analgesic acts by a different mechanism) to improve analgesic efficacy [27,30]. Adjuvants are recommended in neuropathic pain and in patients with associated depression. Serotonin and norepinephrine reuptake inhibitors are a good option since they would have analgesic effects superior to those of traditional selective serotonin reuptake inhibitors [32].

### 4.2. Anorexia/Cachexia

Cancer cachexia is a multifactorial syndrome defined by an ongoing loss of skeletal muscle mass that cannot be reversed by conventional nutritional support and leads to progressive functional impairment [33]. It is associated with an increase in chemotherapy toxicity, increased mortality, and QoL impairment [33]. The presence of anorexia/loss of appetite is associated with an increased risk of malnutrition, sarcopenia, care requirements, hospitalization, falls, and impaired cognition [34]. Anorexia is difficult to control, with the first step being the identification and treatment of issues which may interfere with appetite, such as a dry mouth, pain, or nausea [31].

No widely approved drug for the treatment of cancer cachexia is available. Neither glucocorticoids nor megestrol acetate improve cancer anorexia-cachexia syndrome beyond a few weeks, and may cause AEs, particularly among older adults [35,36]. Promising drugs include ghrelin secretagogues, such as anamorelin, which significantly stimulate appetite in patients with cancer; however, in patients aged ≥65 years, the effects of these drugs are less significant, and their use does not lead to improved handgrip strength or decreased mortality and/or disability [37,38].

In older patients with cancer, the best strategy to ameliorate cachexia can be a multimodal approach including exercise, nutritional support and, in some cases, medications. While these recommendations are part of guidelines issued by the European Society of Medical Oncology (ESMO) and the American Society of Clinical Oncology (ASCO), there is limited evidence regarding their use in older individuals [39,40].

The single-arm NEXTAC-ONE trial examined a multimodal intervention (exercise and nutrition for eight weeks), in 30 patients aged ≥70 years with advanced lung or pancreatic cancer, demonstrating feasibility and low rate of adverse effects [41]. The NEXTAC-TWO study is underway, which is a randomized clinical trial aimed at determining the efficacy of this intervention [42]. In the ENeRgy clinical trial, 45 patients aged ≥65 years with advanced cancer were enrolled and randomized to a personalized program based on exercise and nutrition or standard care for eight weeks [43]. The trial demonstrated feasibility, but was not powered to assess effects on nutritional, functional or QoL outcomes. Qualitative findings demonstrated the positive impact of the intervention, so the ENeRgise study is under development and aims to improve statistical power by recruiting a larger number of patients [43].

### 4.3. Dyspnea

Dyspnea is a subjective experience of respiratory discomfort consisting of qualitatively distinct sensations that vary in intensity [44]. Patients with reversible causes of dyspnea, such as pleural effusion, infection, exacerbation of pulmonary disease, pulmonary embolism, or treatment-induced pneumonitis, should receive targeted treatments [45,46].

Systemic opioids should be offered when non-pharmacologic interventions fail to relieve dyspnea [45,46,47]. Although there is a lack of information on the use of opioids as a treatment for dyspnea in older adults, data suggest that doses required are usually lower than those for treating pain. Since dyspnea is associated with increased anxiety, relaxation techniques and breathing retraining should be performed [31]. If this is not enough, short-acting benzodiazepines can be offered in combination with opioids [48], bearing in mind that these drugs increase the risk of falls, contributing to mortality and morbidity [49]. Non-pharmacologic alternatives should always be considered, limiting opioid and benzodiazepine use to the lowest dose and shortest possible duration.

### 4.4. Delirium

Delirium is a fluctuating disturbance in attention and awareness that represents a decline from baseline status, accompanied by cognitive dysfunction [31]. Delirium incidence ranges from 3% to 45% among inpatients with cancer, increasing to 59% to 88% near the end-of-life [50]. The risk of delirium increases with age [51,52], and its presence is associated with increased morbidity, mortality, length of stay, and need for long-term care [53]. According to psychomotor characteristics and level of arousal, delirium is divided into three subtypes: hyperactive, hypoactive, and mixed type [54]. The most common type of delirium in patients with advanced cancer requiring hospitalization due to poor symptom control is hypoactive delirium, which is the most difficult to diagnose and the hardest to treat, with a mortality of up to 81% during hospitalization (compared with 14% for hyperactive delirium) [55]. The etiology of delirium is multifactorial, ranging from infections to medication-related AEs. In patients with cancer, the possibility of central nervous system metastases or AEs of cancer treatment, including immunotherapy, should always be considered [55,56].

Delirium may interfere with the diagnosis and management of other symptoms, such as pain, which in turn may worsen delirium [31]. It also interferes with the patient’s decision-making capacity, making shared decision-making significantly more complex [57]. The main treatment consists of non-pharmacological interventions focused on prevention [58,59]. However, when non-pharmacological interventions are insufficient, treatment with psychotropic drugs such as antipsychotics medications may be necessary [56].

### 4.5. Nausea

The etiology of nausea in patients with cancer is not limited to AEs of treatment, and other causes such as opioid toxicity, bowel obstruction, or constipation must be excluded [31]. Other causes of nausea may be particularly relevant in older adults, since data show they have a lower risk of chemotherapy-induced nausea and vomiting (CINV) than their younger counterparts [60]. In addition, older adults are more likely to receive reduced chemotherapy dosing, which may also lead to reduced risk of CINV.

None of the current antiemetic guidelines include specific recommendations for older patients receiving chemotherapy, since there is little evidence for it’s use in older individuals [61,62,63]. It seems reasonable to deprescribe or adjust the dosing of antiemetics in older adults with cancer, although this needs to be investigated in controlled trials. Single-dose dexamethasone, for example, has the same efficacy as multiple doses in older adults (excluding cisplatin-containing schedules) [64,65]. Figure 2 illustrates recommendations for prescribing antiemetics in older patients with cancer [60].

## 5. Shared Decision Making in Older Adults with Cancer Receiving Palliative Care

Decision making is an essential component of patient care in every stage of the cancer continuum but can be particularly complex in older adults with advanced cancer. Factors such as age, comorbidities, and functional status need to be balanced with the patient’s goals and values as well as with treatment and prognostic expectations.

Exploring the patient’s priorities regarding preferred health outcomes and identifying what matters most is a core element of both geriatrics and palliative care. The process of considering both the personal preferences, priorities, and goals of the older adult and the professional experience of the health care provider is called shared decision making (SDM) [66]. SDM has been identified by patients as a facilitator for optimal palliative care [66] and there is evidence that it can lead to better outcomes [67,68]. Performing a comprehensive GA can also aid in decision-making in older adults with cancer, and is an established recommendation in international guidelines [69,70]. However, up to 70% of oncologists do not use formal GA to inform treatment decisions for any of their older patients [71]. Screening for depression, cognitive impairment, and addressing burden of care (including mobility and social networks) are particularly important components of a GA that can influence decisions regarding cancer treatment [72].

In geriatric palliative care, family members and caregivers greatly influence the SDM process. Older adults depend more on family members and are more likely to consider their input when making decisions [73]. Involving family members in SDM leads to better understanding of cancer-related information, higher treatment adherence, and greater satisfaction with care [74,75]. Assessing illness understanding and prognostic awareness of patients and their caregivers is essential, since 50–73% of older adults with advanced cancer either have poor prognostic understanding, or their prognostic understanding differs from that of their oncologist [76,77]. Poor prognostic understanding has been associated with receiving more aggressive care at end-of-life and lower utilization of hospice services [78,79]. Palliative care is one of the few interventions that improves illness understanding [80].

### 5.1. Advanced Care Planning in Older Adults

The preferences of older patients with cancer are frequently ignored and they are often considered as lacking capacity for decision making merely because of their age and disease condition [81]. However, it is estimated that 45–70% of older adults cannot make decisions for themselves when approaching end-of-life [82], making SDM more challenging. Some of these challenges can be met by advance care planning (ACP) [19]. ACP is a comprehensive communication approach that ensures adequate documentation and implementation of patient preferences [83]. ACP provides patients and their surrogates the chance to discuss their personal values, life goals, and priorities regarding future medical care, principally in the context of the end-of-life. Many people also decide to put their preferences in writing by completing legal documents called advanced directives or advanced statements, with terms and legislation varying between countries. When compared with younger individuals, older adults with life-limiting illnesses are less often willing to sacrifice quality in exchange for quantity of life [84]. In this setting, the primary goal of ACP is to assist people in receiving care that is consistent with their personal values, goals, and preferences, promoting a more person-centered care.

ACP is associated with greater use of supportive care, reductions in aggressive medical interventions, reduced hospital admission rates, better communication, higher satisfaction with decision-making, improvements in QoL, less anxiety and depression, and reduced cost of healthcare at the end-of-life [85,86,87,88]. Various studies demonstrating the benefit of ACP have been conducted either in older adults or in patients with cancer, but research focused specifically on older adults with cancer is lacking.

### 5.2. Cognitive Impairment and Shared Decision Making

Decision-making among older adults with cancer can be affected by the coexistence of dementia which, like cancer, becomes more prevalent with age [89]. The prevalence of dementia is estimated to be between 3.8% and 7% among older adults with cancer [90]; however, it is probably underestimated due to underdiagnosis of cancer in people with dementia and of dementia in people with cancer [91]. Cognitive impairment may also be more relevant in end-of-life care, since up to 54% of patients enrolled into hospice services can have previously undetected cognitive impairment when undergoing a comprehensive neuropsychological assessment battery [92].

Patients with cancer and dementia have greater healthcare needs and poorer clinical outcomes compared with patients with cancer without dementia [93,94]. Coexistence of these diagnoses also complicates the SDM process in several ways, mainly by impairing decision-making capacity (DMC), limiting the ability of patients to provide consent for treatment, and creating communication difficulties among patients, caregivers, and clinicians [95]. Assessing DMC is fundamental for older adults with cancer given the increased prevalence of cognitive impairment, although cognitive impairment by itself is not enough to consider a patient incapacitated [84]. Cognitive decline may influence capacity depending on the degree to which cognition is affected, the underlying cognitive reserve, and the complexity of the situation at hand [96]. When capacity is compromised, patients rely heavily on their caregivers to support decision-making. In advanced cancer, much of the decision-making relates to treatment options and goals in palliative and end-of-life decisions [96,97]. Older adults with dementia or any form of cognitive impairment can greatly benefit from the coordination of palliative care specialists and geriatricians to prioritize their health needs, develop a coordinated plan of care, manage other geriatric syndromes, and optimize their management.

Impaired cognition makes it more difficult and sometimes impossible for a patient to weigh treatment options against their own priorities, representing a challenge in determining their wishes for end-of-life care. A way to assure the patient’s wishes and priorities are considered is to offer ACP early in the dementia trajectory while the patient possesses decisional capacity. This may lead to completion of advanced directives or statements, although these are not legally binding in all countries, and it may be challenging to know whether it applies to the current situation [72]. The most important part of this process is to learn the patient’s priorities, his/her general preferences for health care, and the choice of the substitute decisionmaker. For patients with more advanced dementia who have already lost their DMC, it is up to the substitute decision maker to participate in ACP conversations with the medical team and ultimately make decisions on their behalf. Nonetheless, patients should always have a voice in decisions around their care, even when capacity is absent or decreased.

Patients with dementia are overall less likely to be referred to palliative care [98]. Future research needs to focus on the impact dementia has on cancer treatment decision-making and promote the appropriate usage of palliative care services for this growing population.

## 6. Delivery of Palliative Care for Older Adults with Cancer

### 6.1. Hospitals, Long-Term Care Facilities, and Home Care

Demographic changes have led to an increase in the number of older people who live in long-term facilities or obtain medical care at home, as well as in the number of older caregivers [99]. Most patients living in long-term care facilities are aged ≥80 years, 86% of them have multiple comorbidities (43% with ≥4 chronic conditions) and symptoms, and would benefit from early identification of palliative care needs and comprehensive support [100]. However, long-term care facilities often have difficulty meeting these needs [100,101]. In a US study, older adults with metastatic cancer living in long-term care facilities were more likely to receive aggressive end-of-life care, and less likely to obtain supportive and palliative care than those living in the community [102].

In a survey conducted in seven European countries, between 50 and 83% of respondents stated they would prefer to die at home if they had cancer [103]. However, providing end-of-life care at home represents a challenge for the healthcare system and for unpaid caregivers, many of whom are older adults [104]. Comprehensive GA and multidisciplinary interventions may improve nutrition, QoL, and function in older adults with cancer receiving palliative care at home, although evidence is very limited [105,106,107]. Importantly, interventions aimed at providing supportive care for older adults should prioritize maintaining the QoL of caregivers [106].

### 6.2. Resource-Constrained Settings

In low- and middle-income countries (LMICs), early access to palliative care represents a challenge. In Mexico, for example, <50% of patients with advanced cancer receive supportive care consultations and discuss end-of-life care and ACP in the first year after diagnosis [108]. Among the barriers for accessing geriatric palliative care in LMICs, one of the most important is the lack of resources and personnel [109,110]. The number of geriatricians in most LMICs is far below the number recommended by geriatric societies (one geriatrician per 700 persons aged ≥65 years) [111]. Brazil, for example, despite being the pioneer country in the training of geriatricians in Latin America, has only one geriatrician per 37,000 older adults [112,113].

The development of multidisciplinary teams including geriatrics, oncology, and palliative care is a top priority to face the demographic changes taking place in LMICs [114]. A potential solution to bring these scarce resources closer to vulnerable populations is telemedicine, which has been validated as a useful tool in geriatrics, oncology, and palliative care [115,116,117,118]. Importantly, a key component for advancing global palliative care is the training of nurses. Although several pilot projects in LMICs have successfully integrated nurses into community-based palliative care teams, an important gap is the lack of inclusion of geriatric principles, which is also a significant priority in geriatric oncology [114,119,120].

### 6.3. Transition and Coordination of Care

Patients with cancer aged ≥65 years are less likely to be referred to palliative care than their younger counterparts [121]. Triggers for referral to palliative care are usually geriatric problems: frailty, functional dependence, cognitive impairment, emotional distress, and caregiver-related problems [122]. Early referral is particularly relevant for older individuals, since those receiving integrated geriatric care have a lower risk of unplanned hospitalizations and shorter length of hospital admission [123]. In contrast, older adults with cancer receiving specialized palliative care without input from a geriatrician are at increased risk of inappropriate prescribing, particularly near the end-of-life [124].

## 7. Conclusions

The growing number of older patients worldwide, and particularly in developing regions of the world, will bring along a significant increase in the number of cancer cases and, consequently, in the need for palliative and supportive care. Providing adequate supportive and palliative care for older adults with cancer requires the participation of all stakeholders, including clinicians, researchers, policy makers, civil society, caregivers, and of course, patients. Integrating geriatric competencies in the training of palliative care physicians and nurses, as well as basic palliative care training in the curricula of geriatricians, should be a priority for the future of both disciplines. The nuances of treating older individuals, including specific changes in physiological, social, cognitive, and functional characteristics, should always be evaluated using validated tools, and considered when providing palliative care and when discussing goals of care and ACP. Researchers should also strive to include older patients and their caregivers into studies of novel palliative care interventions to grow the evidence base and generate tailored care delivery strategies. The field of geriatric palliative care must become a basic component of the healthcare system, and an integral part of cancer care to improve the quality of care we provide to a vulnerable and growing population.

## Figures and Tables

**Figure 1 cancers-15-03858-f001:**
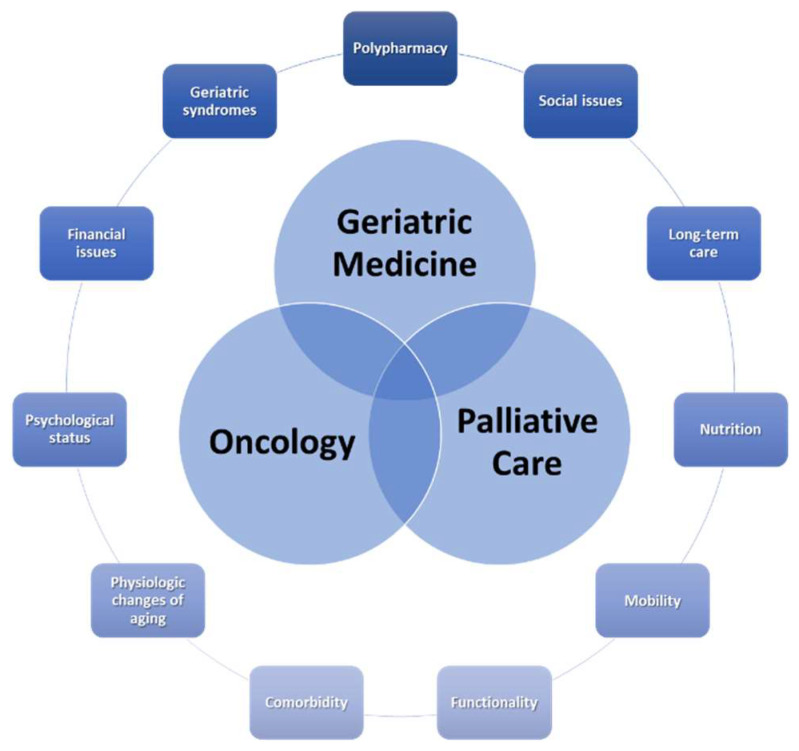
The structure of palliative and supportive care in older adults with cancer.

**Figure 2 cancers-15-03858-f002:**
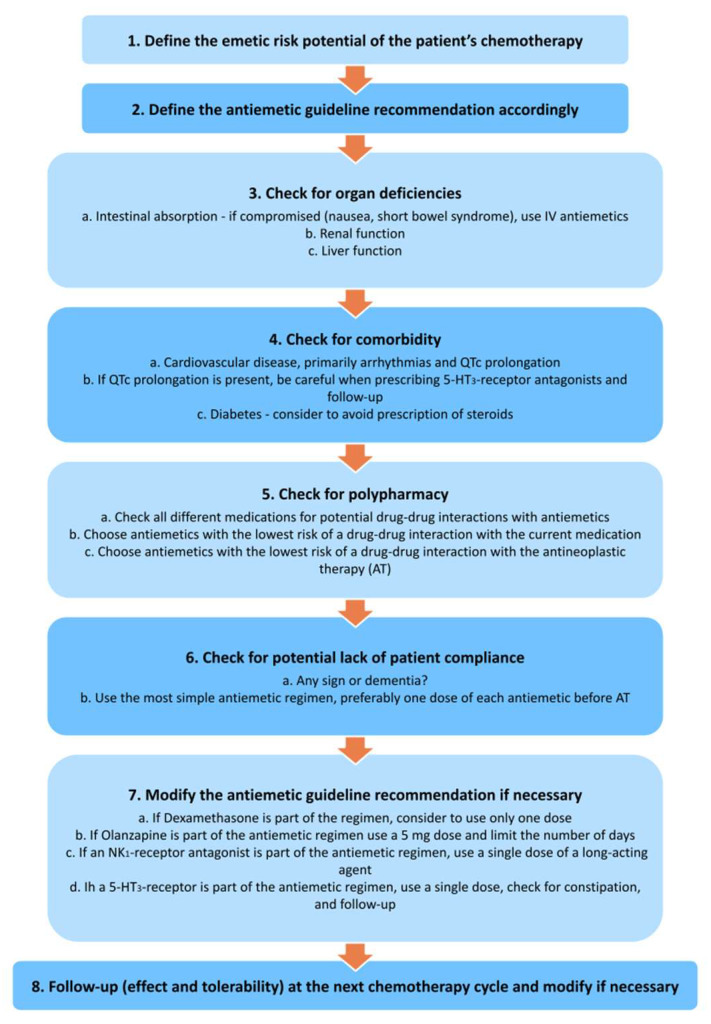
Recommendations for prescribing antiemetics in older patients with cancer [59].

**Table 1 cancers-15-03858-t001:** Proposals for collaboration in common areas for geriatrics and palliative care from the American Geriatrics Society and the American Academy of Hospice and Palliative Medicine [21].

Workforce
1. Continued dialogue on ways to train mid-career providers since fellowship training will not meet workforce needs
2. Better understanding of the current workforce issues in hospice and palliative medicine and geriatrics through incorporation of questions into workforce studies and surveys
3. Identification of areas of resistance to collaboration
4. Delineation of unique and overlapping competencies
**Research**
1. Joint advocacy for research in advanced illness/multimorbidity/symptom burden and symptom management in older adults
2. Increased communication to relevant stakeholders regarding the vacuum in geriatrics/hospice and palliative medicine research
3. Increasing research funding
**Education**
1. Including educational material regarding geriatrics and palliative care in specialized conferences for both specialties
2. Joint memberships in international and national societies
**Policy**
1. Confer on matters of mutual clinical and policy importance
2. Share policy and advocacy initiatives regularly to highlight areas of mutual interest and emphasis
3. Prepare statements and political approaches to “hot button” issues, including but not limited to rationing, healthcare reform, end-of-life care, Medicare cost-cutting initiatives

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
