# Peer review of "Delivering Palliative and Supportive Care for Older Adults with Cancer: Interactions between Palliative Medicine and Geriatrics"

_cancers, 2023, doi:10.3390/cancers15153858_

Round 1
Reviewer 1 Report
My comments to the authors are:
1) This issue is relevant due to the progressive ageing of the population, leading to an increase in the incidence of cancer and, logically, a higher prevalence of cancer in the elderly. The health care and decision-making process for older adults with cancer is complex because of the need to consider their co-morbidities, polypharmacy, frailty, functional status, disability, risk or presence of cognitive impairment, emotional impact, functional status, existential meaning of life, and socio-family support, etc.
2) Medical oncologists are naturally sensitive to older people with cancer, but often have little training in geriatrics. This paper highlights the need for an always individualised assessment, often based on validated tools, to enable decision making and care planning appropriate to the actual condition of the patient. Tailored planning is therefore needed to ensure that no older patient misses the opportunity to receive cancer treatment appropriate to their frailty, that no older patient receives disproportionate cancer treatment, and that all receive appropriate geriatric and palliative care.
3) In line 44, "...will be aged ≥65 years [1]". Reference 1 is insufficient. It is necessary to add an internet link and the last date of its consultation.
4) In lines 58-59. “Palliative Care can improve rates...reduce the cost of treatment”. This statement needs to be referenced.
5) In lines 61-63. “In older patients...tailored care” This statement needs to be referenced.
6) The paragraph between lines 90 and 98 emphasises systematic geriatric assessment as the best tool for assessing older people. In this section, I think that there is a lack of mention of various tools that have been developed to assess frailty in older people with cancer (Groningen Frailty Index, G8..), as well as tools that have been developed for decision making and modification of chemotherapy doses according to frailty (CARG..). Can you add any comments on these tools?
7) In the delirium section, I think it would be interesting to briefly describe the two types of cognitive dysfunction: hyperactive and hypoactive.
8) 8) In line 242. "... with psychotropic medication may be necessary". I think it is too generic to say psychotropic medications. It might be better to say "...antipsychotic drugs" or "...with psychotropic drugs such as antipsychotics". Do you agree?
9) In section “Delivery of palliative care for older adults with cancer” , lines 345-385. I think it might be interesting to add some information about the organisation of the different teams that provide specialist palliative care (Hospital Support Teams, Hospital Palliative Care Units, Palliative Consultant services, Home Care Teams...) or oncogeriatrics units.
Author Response
Please see the attachement

Reviewer 2 Report
The manuscript is a review of areas of common interest to palliative medicine, geriatrics, and oncology
Suggestions are provided to the authors to enhance the focus on collaboration and to strengthen the information regarding geriatric oncology
